# LANGUAGE CONFUSION GATE: LANGUAGE-AWARE DECODING THROUGH MODEL SELF-DISTILLATION

**Collin Zhang**[1,2*]**, Fei Huang**[1]**, Chenhan Yuan**[1]**, Junyang Lin**[1]
[1]Qwen Team, Alibaba Group
[2]Cornell University

## ABSTRACT

Large language models (LLMs) often experience language confusion, which is the unintended mixing of languages during text generation. Current solutions to this problem either necessitate model retraining or cannot differentiate between harmful confusion and acceptable code-switching. This paper introduces the **Language Confusion Gate** (LCG), a lightweight, plug-in solution that filters tokens during decoding without altering the base LLM. The LCG is trained using norm-adjusted self-distillation to predict appropriate language families and apply masking only when needed. Our method is based on the findings that language confusion is infrequent, correct-language tokens are usually among the top predictions, and output token embedding norms are larger for high-resource languages, which biases sampling. When evaluated across various models, including Qwen3, GPT-OSS, Gemma3, Llama3.1, LCG decreases language confusion significantly, often by an order of magnitude, without negatively impacting task performance.

## 1 INTRODUCTION

Large language models have made remarkable strides in multilingual understanding and generation, with state-of-the-art systems like Qwen3 and GPT-5 now supporting over 100 languages and achieving strong performance on benchmarks such as FLORES-200 (Team et al., 2022) and XL-Sum (Hasan et al., 2021). These models demonstrate impressive cross-lingual transfer capabilities, enabling applications ranging from translation to multilingual creative writing. However, despite their sophistication, even the most advanced LLMs occasionally make seemingly elementary errors: generating text that mixes languages inappropriately. This phenomenon, known as **language confusion** (Marchisio et al., 2024), occurs when a model outputs tokens from an unintended language family (e.g., inserting Chinese characters into an Hebrew sentence), undermining reliability and user experience. We show three examples of language confusion mistakes on left side of Figure 1.

While recent improvements have reduced confusion rates in some models, the trend of Large Reasoning Models seem to reintroduce the problem. As discussed in Guo et al. (2025), DeepSeek-R1 exhibited significant language mixing during training, and applying a language consistency reward led to measurable performance degradation, indicating a trade-off between language consistency and reasoning capability. Wang et al. (2025) shows that the reasoning capability of LLM degrades when thinking in low resource languages, which explains why a reward purely based on outcome correctness encourages language confusion.

Further, our evaluation reveals that language confusion remains widespread, even among leading commercial systems. For instance, GPT-5-Chat exhibits 0.57% Chinese/Japanese (CJ) character confusion and 0.67% Latin-script confusion, while Qwen3-235B-A22B-Instruct-2507 suffers 2.27% CJ and 5.07% Latin confusion. These results confirm that language confusion is far from solved and affects both open-source and proprietary models.

The challenge of mitigating language confusion is the lack of an automatic way of evaluation, rule-based detectors struggle in distinguishing *erroneous* mixing from *legitimate* code-switching, a common and often necessary linguistic behavior. In many practical scenarios such as writing English

---

*Work done during internship at Qwen. Corresponding authors emails: Collin Zhang `rz454@cornell.edu`, Fei Huang `feihu.hf@alibaba-inc.com`

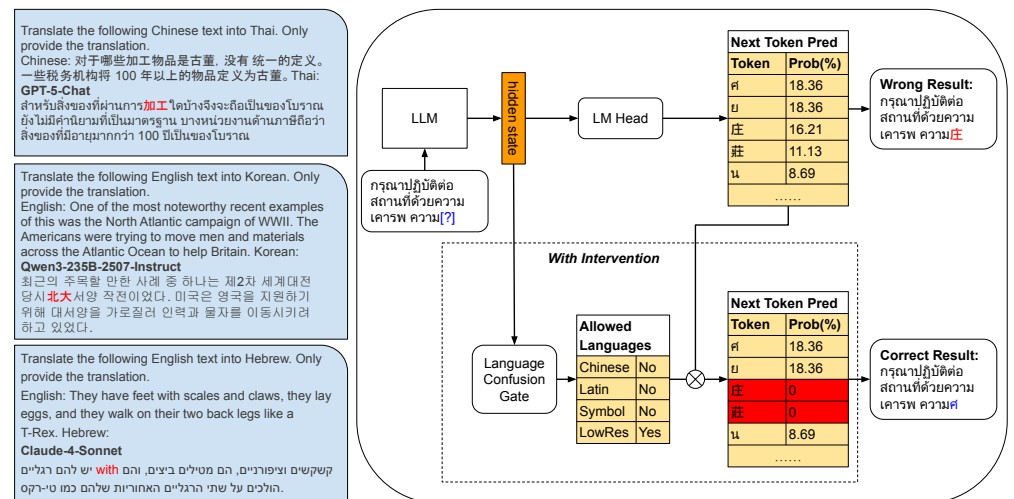

Figure 1: Left: examples of language confusion in three LLMs on the left. Right: LCG takes the LLM's hidden state, predicts permissible language families (CJ, Latin, Symbols, LowRes), and dynamically masks logits for disallowed tokens only when necessary to correct the confusion on the left without altering the base model.

code with Chinese comments, using technical terms like `ReLU` or `Python` in non-English text, or explaining foreign phrases, the ability to blend languages enhances expressivity and utility. Consequently, simply restricting LLM to output in a single language doesn't work. Furthermore, the high number of supported languages requires us to find a method that can be easily scaled to a large number of languages.

To address this, we analyze generation behavior at confusion points and make three key observations: (1) language confusion occurs rarely—suggesting the model generally knows the correct language; (2) at the confusion point, the correct-language token typically ranks within the top-5 candidates, indicating that LLM knows the correct answer; and (3) A mechanistic observation that output layer token embedding norm imbalance makes LLM biased towards high-resource languages.

Leveraging these insights, we propose the **Language Confusion Gate (LCG)**: a lightweight, plug-in intervention that dynamically filters inappropriate tokens at decoding time without modifying the base LLM. LCG consists of a small two-layer MLP trained via self-distillation on the frozen model's own top-$k$/$p$ predictions, with norm adjustment used to debias inclination to high-resource language tokens. At inference, the gate predicts which language families (Chinese/Japanese, Latin, Symbols, Low Resource Languages) are permissible at each step, and applies masking only when necessary.

Results show that LCG reduces language confusion by an order of magnitude across multiple models and tasks. For example, on FLORES-NO-LATIN benchmark, our method reduces Chinese/Japanese confusion from 1.0% to 0.0% and Latin confusion from 4.4% to 0.4% in Qwen3-30B-A3B.

We make three key contributions that build upon our analysis of language confusion in LLMs: **Firstly**, we propose the Language Confusion Gate (LCG), an efficient intervention mechanism that dynamically filters inappropriate tokens during generation without modifying the base LLM. **Second**, we introduce norm-adjusted self-distillation, leveraging mechanistic insights about token embedding norm imbalance to train the gate using the model's own debiased top-k/p predictions. **Third**, we collect and open-source specialized training and evaluation datasets, and evaluate LCG on open-source models covering diverse architectures and both thinking and no-think modes.

## 2 RELATED WORKS

Language confusion has emerged as a critical challenge in multilingual large language models (LLMs). Marchisio et al. (2024) formalized the concept of language confusion and introduced the Language Confusion Benchmark (LCB), providing the first standardized evaluation framework for

measuring cross-lingual interference in LLMs. Their analysis revealed that confusion often occurs at specific "confusion points" in the generation process, motivating targeted intervention strategies. They show that greedy decoding can help reduce language confusion but not eliminate it, at the cost of potentially degraded LLM performance.

Building on this foundation, Nie et al. (2025) conducted mechanistic interpretability analysis to identify neurons responsible for language switching behavior. They found that suppressing these critical neurons during inference reduces unwanted language mixing, suggesting that confusion is localized in the model's internal representations. Similarly, Ji et al. (2025) focused on Korean-language setups where Chinese character intrusion was observed, proposing a post-hoc smoothing method that identifies and suppresses Chinese tokens during decoding. Other approaches have explored different mitigation strategies.

Li et al. (2025) took a unique perspective by studying whether language mixing between English and Chinese could actually benefit reasoning performance. Rather than targeting suppressing language confusion, they trained a gating mechanism to predict when mixing helps or harms task performance, selectively encouraging or discouraging it accordingly. In contrast, Lee et al. (2025) proposed training models to prefer language-consistent responses through Odds Ratio Preference Optimization (ORPO), aligning model outputs with preferences for linguistic coherence.

While these works represent important progress, they face limitations in practical deployment: some require model retraining or fine-tuning, others lack the ability to distinguish legitimate code-switching (e.g., technical terms or bilingual education contexts) from erroneous confusion. Our work addresses these gaps by introducing a lightweight, plug-in intervention that operates at confusion points without modifying the base model, while preserving valid multilingual behaviors.

# 3 CLOSER LOOK INTO LANGUAGE CONFUSION

## 3.1 CONFUSION POINT

Large language models (LLMs) generate text autoregressively by producing a probability distribution over the vocabulary at each step. The next token is usually sampled using a hybrid of top-k and top-p (nucleus) sampling. As demonstrated in Marchisio et al. (2024) and Nie et al. (2025), a **confusion point** arises when a token with a language different from the last token appears within the sampling tokens. We define the token in the different language as **confusion token**.

To better understand behavior of LLMs at confusion points, we use the **FLORES-NO-LATIN** dataset as described in Section 5.2 to trigger language confusion in Qwen3-8B. We inspect the token probability distribution of LLM at the confusion point, and we find that the confusion token is the top-1 token 56.74% of the time, which makes **greedy decoding** ineffective to prevent language confusion. Further, we find that language consistent tokens appear within top-3 99.29% of the time. This suggests that language switching errors are not due to a complete absence of correct-language candidates in the model's output distribution, but rather to the model assigning insufficient probability mass to them relative to competing tokens from the confused language. This observation motivates a logits based intervention strategy without modifying weights of the model. We can simply mask tokens in the undesired language families.

## 3.2 TOKEN EMBEDDING NORM ANALYSIS.

The magnitude of output token embeddings plays a critical role in language confusion by favoring tokens from high-resource languages.

Language models compute hidden states at each generation step and project them to vocabulary-sized logits through a linear layer $W_{out} \in \mathbb{R}^{d_{model} \times |V|}$:

$$\text{logits} = W_{out}^\top h$$

$W_{out}$ is a collection of individual column vectors, $[e_0, e_1, ..., e_{|V|-1}]$, where each vector $e_i$ represents the output embedding for a specific token in the vocabulary, we define these as **output token embeddings**. Then we can decompose $\text{logit}_i$ as

$$\text{logits}_i = h \cdot e_i = ||h|| \cdot ||e_i|| \cdot \text{cos\_sim}(h, e_i).$$

That is, the logit of each token $\text{logit}_i$ is the **dot product** between the hidden state $h$ and that token's embedding $e_i$, and thus be decomposed into its geometric components: magnitude (norm) and direction (cosine similarity).

Since the norm of the hidden state, $||h||$, is constant for all tokens at a given generation step, this decomposition reveals a critical, often-overlooked factor: the output token embedding norm, $||e_i||$. It shows that a token can achieve a high logit simply by having a large embedding norm. This creates a systemic bias where tokens from high-resource languages develop larger norms, which sometimes causes language confusion.

We categorize the vocabulary by language family: CJ (Chinese/Japanese), Latin, and Low-Res (low-resource languages). For each language family, we compute the fraction of tokens whose embedding norms lie in the top 5% of all token norms in the model's vocabulary. As shown in Table 1, the results confirm a significant imbalance: high-resource languages like Latin and CJ consistently dominate the high-norm group, while low-resource languages are heavily underrepresented.

Table 1: Percentage of tokens in each language family with embedding norms among top 5% of all token embedding norms.

| Model | CJ% | Latin% | Low-Res% |
|---|---|---|---|
| Qwen3-8B | 10.74 | 4.61 | 0.14 |
| Qwen3-30B-A3B | 6.52 | 5.50 | 0.07 |
| Llama3.1-8B | 4.38 | 5.95 | 1.34 |
| Gemma3-12B | 0.94 | 5.04 | 2.40 |
| GPT-OSS | 0.00 | 7.00 | 0.00 |

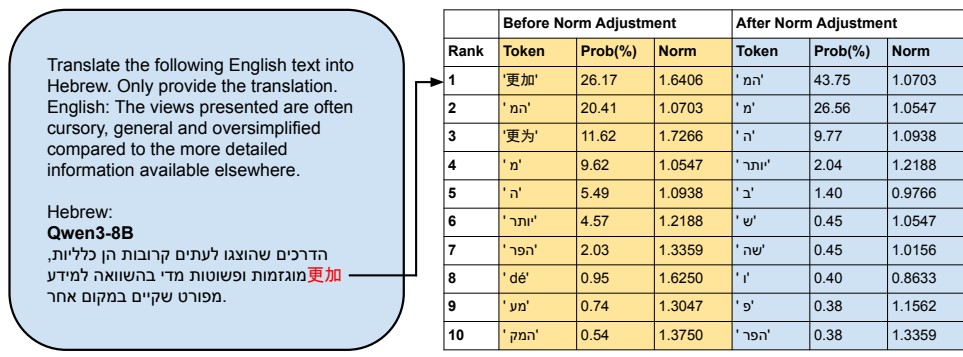

Figure 2: Top-10 Logits before and after applying norm at a confusion point on Qwen3-8B

Adjusting the logits by the token embedding norm: $\text{logit}_{adj,i} = \frac{h \cdot e_i}{||e_i||} = ||h|| \cdot \text{cos\_sim}(h, e_i)$ removes the embedding norm bias, allowing tokens to be ranked purely by their cosine similarity with the hidden state as shown in Figure 2. We can see that the initial highly ranked language confusion tokens disappear from the top 10 tokens. This shows that norm-adjusted top-k tokens provides a signal for correct next-token language family, and we can use this signal to train a gate that predicts language family of next token as discussed in Section 4. Norm bias can account for a subset of such errors but cannot fully explains language confusion. For example, it can't explain language confusion between English and Chinese since they both have high norm, or between low resource languages since they both have low norm, so it can't be directly used for intervention.

We studied how percentage of low resource tokens in training data affects the output token norm in Appendix G.

### 3.3 LANGUAGE CONFUSION V.S. NATURAL LANGUAGE MIX

Language mix, or code-switch, has been observed and discussed in both NLP and linguistic area (Doğruöz et al., 2021; Winata et al., 2023). We show several examples that language mixing in the context is necessary. 1) the use of English abbreviations or terms like Python, Java, iPhone. 2) Coding tasks. The user may prompt in Chinese to ask LLM to write code, while most programming languages are based on English characters. 3) Language study. We show examples of natural code-switch contexts in Appendix H. The user may ask the language model to explain phrases in another language in English. In that case, the ability to use several language in a response should be preserved, so simply constrain the LLM to output in single language won't work. In that case, simply enforcing a rule-based language consistency constraint won't work, since we have two objectives: suppressing unnormal language confusion while maintaining normal code-switch capabilities.

### 3.4 DOES SOTA COMMERCIAL LLMS SHOW LANGUAGE CONFUSION?

Even state-of-the-art commercial LLMs exhibit non-negligible language confusion, confirming it as a widespread challenge across both open-source and proprietary models. We evaluate commercial models using our **FLORES-NO-LATIN** as described in Section 5.2 (Table 2). The results reveal that language confusion is widespread—even among state-of-the-art closed models. Notably, we are not sure that if a similar intervention mechanism like Language Confusion Gate has been applied to any commercial models, but we can observe that all models show non-negligible Latin Confusion and CJ Confusion (except Claude-Sonnet-4). We show the full table covering more Commercial LLM confusion rates in Appendix E.

Table 2: **Language Confusion Rates on the FLORES-NO-LATIN Benchmark for Leading Commercial LLMs.** This table displays the percentage of responses containing erroneous Chinese/Japanese (CJ%) and Latin (Latin%) characters, alongside the task-specific BLEU score. These results highlight that language confusion is a persistent issue across various SOTA models.

| Model | CJ% | Latin% | BLEU |
|---|---|---|---|
| GPT-5-Chat | 0.57 | 0.67 | 10.66 |
| Claude-Sonnet-4 | 0.00 | 0.35 | 21.77 |
| Gemini-2.5-Pro | 0.04 | 0.50 | 19.11 |
| DeepSeek-v3.1 | 0.67 | 1.06 | 18.11 |
| Qwen3-235B-Instruct | 2.27 | 5.07 | 15.43 |

## 4 METHOD

### 4.1 LANGUAGE CONFUSION GATE

We propose a lightweight intervention mechanism to address language confusion without modifying the base LLM architecture or requiring model retraining. Our approach introduces **Language Confusion Gate** as shown in Figure 1: a two layer MLP that determines language families allowed at each generation step, then masks inappropriate tokens from the logits during sampling. The gate itself is a two layer MLP that takes the LLM's final hidden state $\mathbf{h}_t$ as input and produces language family logits $\mathbf{z}_t = \mathrm{MLP}(\mathbf{h}_t) \in \mathbb{R}^4$. At each generation step, for each new token, the gate predicts one or more language families allowed for next token, and mask tokens in banned language. Intervention only happens when the tokens could be sampled under current top-k, top-p, and temperature parameters contain language families disallowed by the gate. Since language confusion happens rarely, it has minimal impact on overall generation.

**Classify Tokens into Language Families.** To enable the Language Confusion Gate, we classify each token in the entire vocabulary into one of four mutually exclusive families: Chinese and Japanese (**CJ**), for tokens primarily composed of Chinese and Japanese characters; **Latin**, for tokens representing Latin scripts; **Symbols**, for punctuation, numbers, and special characters; and Low Resource Languages (**Low-Res**), the category for all other tokens.

The classification is performed using a prioritized heuristic. For each token in the vocabulary, we first attempt to decode it from its byte-pair encoding (BPE) into Unicode characters. If the resulting characters contain any Chinese or Japanese script, the token is classified as **CJ**. If not, and the characters consist only of Latin script and symbols, it is classified as **Latin**. If a token decodes exclusively to symbols, it is classified as **Symbols**. All other tokens that decode to valid characters from other scripts are categorized as **Low-Res**.

A known challenge with BPE is that some tokens may represent incomplete Unicode characters. In these cases, we analyze the partial byte sequence to infer its language family based on Unicode's continuous block structure. We discuss in more detail this method in Appendix A. If the family cannot be reliably determined, the token is conservatively classified as **Symbols**. Applying this methodology to the Qwen3 tokenizer (151,936 total tokens) yields the following distribution: 27,658 **CJ**, 94,666 **Latin**, 10,355 **Symbols**, and 19,257 **Low-Res** tokens.

## 4.2 TRAINING

**Norm-adjusted self-distillation.** We train the gate with norm-adjusted self-distillation, use the model's own language prediction as pseudo-targets, and remove the systemic advantage of high-norm tokens with norm-adjustment.

For the logit vector $\mathbf{logits} \in \mathbb{R}^{|V|}$ produced at a given step, we compute **norm-adjusted logits**, $\mathbf{logits}_{\text{adjust}}$, by dividing each token's logit by the norm of its output embedding vector $||\mathbf{e}_v||_2$.

With these debiased logits, we create multi-label pseudo-targets $\mathbf{y}_t^*$ for each generation step $t$. First, we identify a set of high-confidence candidate tokens, $S_{k,p}(\mathbf{logits}_{\text{adjust}})$, by applying top-k/top-p filtering to the *norm-adjusted* logits. Then, we determine which language families are present in this candidate set. The pseudo-target for language family $i$ is set to 1 if any token from that family appears in the set, and 0 otherwise. This is formally expressed as:

$$y_{t,i}^* = \mathbf{1}\big[S_{k,p}(\mathbf{logits}_{\text{adjust}}) \cap \mathcal{F}_i \neq \emptyset\big],$$

where $\mathcal{F}_i$ is the set of tokens belonging to language family $i$ (as defined in Section 4.1).

We train the gate to predict the pseudo-targets using a standard binary cross-entropy (BCE) loss: $\mathcal{L} = \sum_{i=1}^{n} \text{BCE}(y_{t,i}^*, \sigma(z_{t,i}))$, where $\sigma$ is the sigmoid function and $n$ is the number of language families. We freeze weights of the LLM during training.

## 4.3 INTERVENTION RULES

During inference, we apply the LCG to dynamically mask tokens from disallowed language families at each generation step. To mitigate potential side effects, we supplement the gate's prediction with several intervention rules: (1) **Symbols and Low-Res tokens are never masked.** It's very rare for high-resource language to mix low-resource languages, so we never mask Low-Res tokens. We never mask symbols since they don't cause language confusion. (2) **No intervention if the gate's prediction is contradicted by high-confidence model output.** If neither of the two high-probability candidate sets defined by (top-$k = 5$, top-$p = 0.999$) or (top-$k = 20$, top-$p = 0.95$) contains any token from the gate-predicted language family, we refrain from applying any mask. (3) **Persistence of the previous token's language.** To maintain linguistic coherence, we always allow the language family of the immediately preceding non-symbol token.

## 5 EXPERIMENTS

## 5.1 EXPERIMENTAL SETUP

**Models.** Our experiments include both standard ("no-think") and reasoning-focused ("thinking") large language models to ensure the LCG is effective across different architectures and capabilities. For no-think models, we applied our intervention to **Qwen3-30B-A3B-Instruct-2507** (Yang et al., 2025), **Qwen3-8B**, **Llama 3.1-8B** (Dubey et al., 2024), and **Gemma3-12B** (Team et al., 2025). For thinking models, we evaluated our intervention on **Qwen3-30B-A3B-Thinking-2507**, **Qwen3-8B**, and **GPT-OSS-20B** (OpenAI, 2025). Notably, **Qwen3-8B** is a hybrid model and was used in both experimental setups. In experiments, we refer the gate trained with norm-adjusted self-distillation

**LCG-adjusted**, while the gate trained only with self-distillation without norm-adjustment **LCG-unadjusted**.

**Training Data for the Gate.**    We trained the LCG on a composite dataset of approximately **78,000 samples** covering over **200 languages** to ensure it learns to handle a wide variety of linguistic contexts. This same dataset was used to train the gate for both thinking and no-think models. The data was aggregated from several sources, including the **Aya Dataset** (Singh et al., 2024) for diverse topics, the **FLORES+ Dataset** (NLLB Team et al., 2024) to generate translation pairs for low-resource languages, the **DeepSeek Distill Dataset** (Lightblue KK., 2024) for multilingual reasoning contexts, and the **Alpaca** (Taori et al., 2023) datasets (Chinese & English) to maintain strong performance in high-resource languages.

## 5.2    EVALUATION STRATEGY

Our evaluation is designed to confirm that LCG reduces language confusion without degrading task performance. We use different benchmarks for "thinking" and "no-think" models to align with their distinct behavioral patterns.

**Evaluation Datasets.**    We evaluate no-think models using the translation dataset **FLORES+** for Arabic, Hebrew, Korean, and Thai, and the **INCLUDE** benchmark (Romanou et al., 2024), a multilingual knowledge and reasoning dataset for Arabic, Hebrew, Greek, Russian, and Vietnamese. For thinking models, we use Python problems from **Humaneval-XL** (Peng et al., 2024) in Arabic and Hebrew, repeating each prompt 10 times to reliably detect confusion in reasoning-intensive tasks. Across all datasets, we measure both language confusion rate and standard task performance.

**Evaluation Metrics.**    We define the language confusion rate as the percentage of model responses that contain at least one character from an unintended language script. Our evaluation focuses on two primary types of confusion: Chinese/Japanese (CJ) confusion and Latin confusion. CJ confusion is straightforward to measure using a rule-based detector, as legitimate code-switching into Chinese or Japanese characters is exceedingly rare in the target languages of our benchmarks. Consequently, we evaluate CJ confusion across all datasets.

In contrast, Latin confusion presents a more nuanced challenge due to the frequent and valid use of Latin-script tokens in contexts such as programming code or mathematical notation. To address this, we partition the FLORES+ dataset into two subsets: **FLORES-NO-LATIN**: translations whose ground-truth references contain no Latin characters, so any Latin script in model output is considered erroneous. **FLORES-WITH-LATIN**: translations where Latin characters appear in the reference and are thus permissible. This partitioning is performed by examining ground-truth translations from English into five target languages: Arabic, Hebrew, Korean, Thai, and Chinese, and flagging those that include Latin-script characters. We restrict our Latin confusion evaluation to the **FLORES-NO-LATIN** subset, where rule-based detection reliably identifies unintended language mixing.

**Rationale for Not Using LCB.**    We use established multilingual benchmarks for our evaluation instead of the Language Confusion Benchmark (LCB) (Marchisio et al., 2024) for two reasons: (1) We observed that some LCB queries require natural code-switching, which could lead to unreliable confusion metrics. (2) Its language detector sometimes produce wrong results, which may result in false positives. Our methodology of using standard benchmarks with targeted filtering provides a more robust and practical evaluation.

## 5.3    EVALUATION RESULTS

We evaluate the Language Confusion Gate (LCG) across both standard ("no-think") and reasoning-focused ("thinking") models to assess its effectiveness in mitigating unintended language mixing and validate the importance of norm adjustment. To examine LCG's impact on legitimate code-switching behavior, we conduct comparative analysis against reference models. Furthermore, we benchmark LCG against established baseline methods to demonstrate its advantages in reducing language confusion while maintaining appropriate multilingual capabilities.

Table 3: **Effectiveness of LCG Intervention on "No-Think" Models. No LCG** is the case without intervention. BLEU scores are for FLORES-NO-LATIN; accuracy is for INCLUDE.

| | Qwen3-30B | Llama3.1-8B | Gemma3-12B | Qwen3-8B |
|---|---|---|---|---|
| **FLORES-NO-LATIN** | | | | |
| CJ% (No LCG) | 1.0 | 3.0 | 0.2 | 4.5 |
| CJ% (LCG-unadjusted) | 0.2 | 2.0 | 0.1 | 0.5 |
| CJ% (LCG-adjusted) | **0.0** | **0.4** | 0.1 | **0.1** |
| Latin% (No LCG) | 4.4 | 8.4 | 1.0 | 12.1 |
| Latin% (LCG-unadjusted) | 0.7 | 5.7 | 0.6 | 6.2 |
| Latin% (LCG-adjusted) | **0.4** | **2.9** | **0.5** | **2.0** |
| BLEU (No LCG) | 13.2 | 11.3 | 16.9 | 12.1 |
| BLEU (LCG-unadjusted) | 13.3 | 12.2 | 17.0 | 11.9 |
| BLEU (LCG-adjusted) | **13.4** | **12.3** | **17.1** | **12.1** |
| **INCLUDE** | | | | |
| CJ% (No LCG) | 2.21 | 0.87 | 0.00 | 1.67 |
| CJ% (LCG-unadjusted) | 0.22 | 0.51 | 0.00 | 0.44 |
| CJ% (LCG-adjusted) | **0.11** | **0.07** | 0.00 | **0.18** |
| Accuracy (No LCG) | 71.12 | 46.12 | 64.95 | 61.43 |
| Accuracy (LCG-unadjusted) | **71.55** | 46.12 | 65.02 | **62.84** |
| Accuracy (LCG-adjusted) | 70.83 | **46.34** | **65.75** | 61.76 |

**Experiments on No-Think Models Intervention.** LCG drastically reduces language confusion in standard (no-think) models by an order of magnitude, while maintaining task performance. We evaluate the intervention effectiveness of the language confusion gate on nothink models on the FLORES+ dataset and the INCLUDE dataset. On the FLORES+ dataset (Table 3), the gate drastically reduces both CJ and Latin confusion. For example, Qwen3-30B-A3B-2507 reduces CJ confusion from 1.0% to 0.0%, and Latin confusion from 4.4% to 0.4%, while maintaining stable BLEU scores. Llama3.1-8B and Qwen3-8B show high language confusion rate without intervention, with LCG-adjusted intervention, Llama3.1-8B's Latin% drops from 8.4% to 2.9% and Qwen3-8B's Latin% falls from 12.1% to 2.0%. Results on the INCLUDE dataset (Table 3) also show significant reductions in CJ confusion—from 2.21% to 0.11% in Qwen3-30B without degradation in task accuracy.

**Frequency of Intervention.** The intervention performed by LCG is sparse and precise, so it has minimum influence on normal token generations. We measured the frequency of intervention for Qwen3-8B and Llama3.1-8B on the FLORES-NO-LATIN dataset. For Qwen3-8B, the intervention rate is 0.38%, which is 523 among 139354 tokens generated. For Llama3.1-8B, the intervention rate is 0.33%, which is 545 among 162846 tokens generated.

**Ablation of norm-adjustment.** We compare the LCG-unadjusted against the LCG-adjusted and find that LCG-adjusted consistently achieves better performance. As shown in Table 3, LCG-adjusted further reduces both CJK and Latin confusion while preserving or slightly improving BLEU scores. For instance, on Llama3.1-8B, Latin confusion drops from 5.7% (LCG-unadjusted) to 2.9% (LCG-adjusted), and on Qwen3-30B, it decreases from 0.7% to 0.4%. This demonstrates that training with norm-adjustment produces a gate with higher accuracy, leading to more precise suppression of language confusion. Thus, **LCG-adjusted** represents our final, optimized intervention.

**Experiments on Thinking Model Intervention.** LCG can also effectively reduce language confusion on thinking models. For reasoning-capable models, we evaluate on the humaneval-xl dataset (Table 4). Our intervention successfully eliminates Chinese character confusion—reducing it from 0.38% to 0.06% in GPT-OSS and from 0.12% to 0.00% in Qwen3-30B—while maintaining competitive Pass@1 and Pass@10 scores. This indicates that the gate effectively prevents language confusion during complex reasoning tasks without degrading the model's reasoning capability, its effect on reasoning length is also very small.

Table 4: Effectiveness of LCG Intervention on "No-Think" Models measured on Humaneval-XL. Length refers to the average Length of reasoning tokens used.

| Model | No LCG | | | | LCG-adjusted | | | |
|---|---|---|---|---|---|---|---|---|
| | CJ% | Pass@1 | Pass@10 | Length | CJ% | Pass@1 | Pass@10 | Length |
| Qwen3-8B | 1.50 | 83.81 | 97.01 | 3327 | 0.06 | 83.13 | 96.67 | 3361 |
| Qwen3-30B | 0.12 | 91.25 | 97.83 | 2528 | 0.00 | 90.50 | 97.97 | 2534 |
| GPT-Oss | 0.38 | 85.88 | 98.07 | 501 | 0.06 | 84.56 | 98.32 | 507 |

**Impact on normal code-switch.** A critical challenge in mitigating language confusion is ensuring that the intervention does not penalize legitimate code-switching, which is a natural and often necessary aspect of multilingual communication. We find that although LCG reduces the frequency of legitimate code-switching, it preserves the model's code-switch ability. We measured LCG's impact on the **FLORES-WITH-LATIN** dataset, a subset of the FLORES benchmark where ground-truth translations contain Latin characters, indicating possibility of code-switch.

In our first experiment, we ran the Qwen3-8B No LCG on the FLORES-WITH-LATIN dataset to generate translations. From these outputs, we select cases where the model's use of English was judged by human annotators to be natural, appropriate code-switch. We then applied Qwen3-8B LCG-adjusted on to these outputs to determine whether it would permit the English tokens at each confusion point. We find that Qwen3-8B LCG-adjusted allows the English code-switch in 86.7% of these human-validated examples, indicating that it largely preserves legitimate code-switch.

In our second experiment, we ran the models with LCG on the FLORES-WITH-LATIN dataset. We define the "code-switch rate" as the percentage of responses that contain Latin characters. We compare the models' rates before and after intervention to two baselines: the rate in the ground-truth answers ("Answer Rate") and the rate of a strong baseline model: Claude Sonnet 4. Notice that these two baselines are just references for comparison but not a ground truth optimal code-switch rate.

As shown in Table 5, LCG does reduce the rate of code-switching across all models. For instance, the code-switch rate for Qwen3-8B from 46.34% to 25.90%. However, the post-intervention rates remain higher than the Claude Sonnet 4 baseline (23.29%) and not much lower than the ground-truth answer rate (38.36%). This suggests that while LCG makes models more cautious about mixing languages, it does not eliminate their ability to perform necessary code-switching. The intervention effectively moderates the behavior, preserving the model's capacity for legitimate language blending while suppressing erroneous confusion. We show examples that our LCG avoids language confusion and maintains natural code-switch in Appendix I.

Table 5: Impact of LCG on legitimate code-switching behavior. In addition to these numbers, at the token level, LCG-adjusted allows English tokens at 86.7% of the confusion points.

| Model | No LCG | LCG-adjusted | Answer Rate | Claude Sonnet 4 |
|---|---|---|---|---|
| Llama3.1-8B | 42.51 | 31.60 | 38.36 | 23.29 |
| Qwen3-8B | 46.34 | 25.90 | 38.36 | 23.29 |
| Gemma3-12B | 30.94 | 25.57 | 38.36 | 23.29 |

**Comparison with baseline intervention mechanisms.** We compared our LCG-adjusted approach with three baseline intervention mechanisms: in-context learning (ICL), greedy decoding, and ORPO tuning as described in Lee et al. (2025). We show the prompt we used for ICL in Appendix D. For the ORPO method, we prepare a multilingual dataset, and synthesize samples with language confusion as rejected samples similar as Lee et al. (2025). The results in Figure 3 demonstrate that, LCG most effectively reduce the language confusion rate while preserving model performance.

For instance, with the Qwen3-8B model, ICL only offers a marginal improvement, reducing the Chinese/Japanese character confusion (CJ%) from 4.5% to 4.2%. Greedy decoding provides similarly limited benefits, lowering the CJ% to just 4.2%. Since greedy decoding is the most conservative sampling strategy, this result implies that merely tuning other decoding parameters like temperature

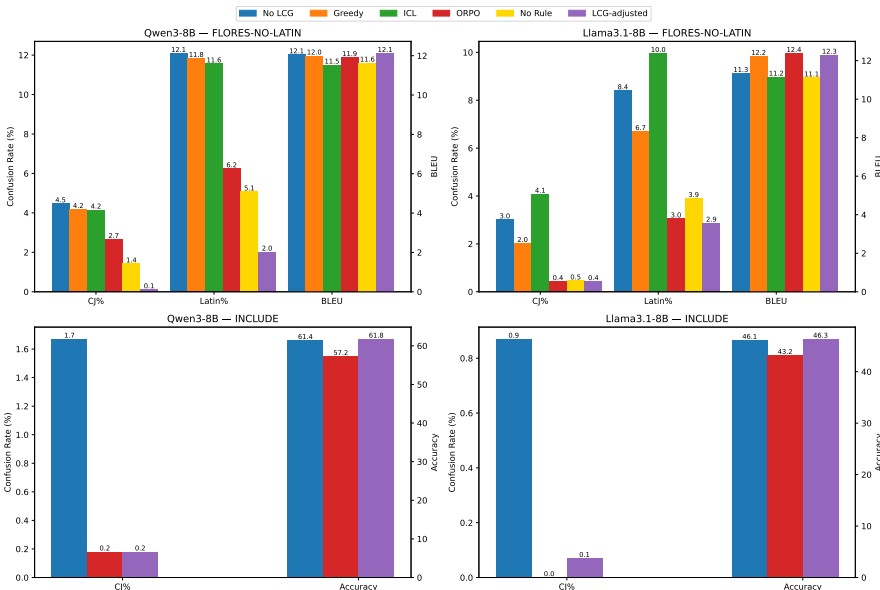

Figure 3: Comparison of LCG with baseline methods. CJ% and Latin% show CJ confusion rate and Latin confusion rate respectively.

or top-p would also be insufficient to resolve the language mixing issue. In contrast, our LCG-adjusted mechanism achieves a substantial and consistent reduction in errors across all models. For Qwen3-8B, it decreased the CJ% from 4.5% down to 0.1% and the Latin% from 12.1% to 2.0%. This shows our learned gate is a more targeted and effective solution than simple prompting or decoding-based interventions.

LCG also outperforms training-based methods. For instance, while ORPO achieves performance comparable to LCG on Llama3.1-8B, it performs significantly worse on Qwen3-8B when evaluated on the FLORES-NO-LATIN dataset. Moreover, we observe that ORPO can degrade the model's general capabilities: on Qwen3-8B, INCLUDE accuracy drops from 61.4 to 57.3, and on Llama3.1-8B, it declines from 46.1 to 43.2. This suggests that ORPO may sacrifice overall language understanding in its attempt to reduce language confusion. We have also ablated the intervention rules as discussed Section 4.3 in the "No Rule" setup. We can see that LCG can still reduce language confusion without the additional rules, but the combination of rules and LCG further reduces language confusion rate. This shows the necessity of both LCG and intervention rules.

## 6 DISCUSSION AND CONCLUSION

The **Language Confusion Gate (LCG)** is a lightweight, plug-in intervention that effectively mitigates language confusion without altering the base model's parameters. Its primary advantage is its practicality: as a small MLP with a sparse intervention rate, it adds minimal computational overhead and avoids the performance trade-offs common in methods that require retraining.

LCG is very efficient. In our production system, we benchmarked the performance of Qwen3-30B-A3B-Instruct 2507 with and without LCG. We set the input length to 2000 tokens with a concurrency of 8 samples, and let the model outputs 100 tokens. We find the time taken for each generation step is 15.95ms without the gate, and 15.99 with the gate, with a minimum increase of 0.4%. Further, LCG is compatible with speculative decoding as discussed in Appendix F.

However, the current approach is limited by its script-level granularity. By grouping tokens into broad families like "Latin" or "Low-Res", the gate cannot resolve more nuanced confusion between languages that share the same script (e.g., Spanish vs. English) or between two different low-resource languages. Future work could explore more fine-grained and language-specific gates.

## REPRODUCIBILITY STATEMENT

To ensure the reproducibility of our results, we have uploaded all code necessary for training and evaluating the models described in this paper to the supplementary materials. The provided code-base includes detailed instructions for data preprocessing, model training, hyperparameter settings, and evaluation procedures. All experiments can be replicated using the included scripts and configurations.

## LLM USAGE

The authors acknowledge the use of large language model (LLM) technology to assist in the preparation of this manuscript. Specifically, an LLM was employed to aid in refining language, improving clarity, and polishing the prose of certain sections.

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

## A  DETAILS OF THE TOKEN CLASSIFICATION ALGORITHMS

A known challenge with Byte-Pair Encoding (BPE) is that some tokens may represent incomplete Unicode characters, making them difficult to classify directly. When a BPE token cannot be decoded into a valid Unicode string, we employ a more sophisticated method to infer its language family by analyzing its underlying byte structure. This approach allows us to conservatively classify even partial tokens.

Our methodology relies on the continuous block structure of the Unicode standard. Even if a token only contains the first few bytes of a multi-byte character, those bytes can significantly narrow down the range of possible code points it could belong to. The process is as follows:

1. **Byte Sequence Decomposition**. First, the ambiguous token is converted back into its raw byte sequence. This sequence is then fed into a UTF-8 state machine that breaks it down into logically complete or partial UTF-8 units. The key is to identify what we term a **"left partial"** token: a byte sequence that correctly starts a multi-byte character but is incomplete.

2. **Inferring Code Point Bounds**. With a "left partial" sequence identified, its first two bytes are used to determine the possible range of Unicode code points it could represent if completed. Based on the rules of UTF-8 encoding, we calculate the lowest and highest possible code points that can be formed. For a 3-byte sequence, the final code point is formed from bit patterns across all three bytes. Since the third byte is missing from our partial token, we can calculate the full range of possibilities by assuming the third byte is at its minimum valid value (`0x80`) and then its maximum valid value (`0xBF`).

3. **Overlapping with Language Blocks**. Finally, we check if this inferred Unicode range overlaps with any predefined language blocks. The calculated bounds are compared against a map of known Unicode ranges, which contains the official start and end code points for scripts like CJK Ideographs, Hiragana, and Katakana.

If a token is identified as a "left partial" and its inferred Unicode range overlaps with any of the CJK blocks, it is classified as a **CJ** token. This robust, byte-level analysis allows us to handle the ambiguity of partial BPE tokens and improve the accuracy of our language family classification.

## B  MORE EVALUATION ON IMPACT ON GENERAL CAPABILITIES

Beyond the benchmarks we evaluated in Section 5, we also evaluated and compared the performance of Qwen3-30B-A3B-Instruct-2507 with and without the Language Confusion Gate in Table B. We can see that the incorporation of LCG doesn't affect general Capabilities of LLM.

| Benchmark | Qwen3-30B No LCG | Qwen3-30B LCG |
|---|---|---|
| MMLU-ProX-13 | 71.95 | 71.74 |
| MMLU | 85.04 | 85.03 |
| MMLU-pro | 78.44 | 78.25 |
| GPQA | 70.35 | 70.45 |
| SuperGPQA | 53.38 | 53.24 |
| AIME24 | 74.32 | 74.90 |
| AIME25 | 61.25 | 61.88 |

Table 6: Performance comparison between Qwen3-30B-A3B-Instruct-2507 without and with LCG.

## C  COMPUTE RESOURCES REQUIRED FOR EXPERIMENTS

All of the training and evaluation experiments we talked about could be run on two NVIDIA-A100 GPUs, for which smaller models like Qwen3-8B, Llama3.1-8B and Gemma3-12B could be trained on single NVIDIA-A100 GPU. It takes about 12 hours to trained the largest model Qwen3-30B-A3B we experimented with on 2 A100 GPUs.

## D  ICL PROMPT

As discussed in Section 5.3, we tried to use a prompt to teach LLM what is language confusion, and provide an example to let it avoid. We show the ICL prompt we use in Figure 4.

Language confusion in the context of large language models (LLMs) refers to the phenomenon where a model mixes or confuses elements of multiple languages inappropriately during generation. Pay attention to prevent it. For example, "곧 방호복을 입은 경찰관들이 yard에 들어와 타격 가스로 수감자들을 몰아넣었다." is incorrect. It should be "곧 방호복을 입은 경찰관들이 마당에 들어와 최루가스로 수감자들을 몰아넣었다." Pay attention to avoid language confusion during generation.

Figure 4: Prompt we used for ICL as described in Section 5.3

## E  COMMERCIAL LLMS CONFUSION RATE

We show the full list of commercial LLMs we tested with FLORES-NO-LATIN in Table 7.

Table 7: Full list of language confusion rate of commercial chat LLMs

| Model | CJ (%) | Latin (%) | BLEU |
|---|---|---|---|
| GPT-5-Chat | 0.57 | 0.67 | 10.66 |
| GPT-5-mini | 0.04 | 1.06 | 18.07 |
| GPT-5-nano | 0.04 | 1.24 | 16.79 |
| Qwen3-235B-Instruct | 2.27 | 5.07 | 15.43 |
| Qwen3-30B-Instruct | 0.92 | 4.68 | 13.42 |
| Claude-Sonnet-4 | 0.00 | 0.35 | 21.77 |
| Gemini-2.5-Pro | 0.04 | 0.50 | 19.11 |
| Gemini-2.5-Flash | 0.07 | 0.82 | 19.93 |
| DeepSeek-v3.1 | 0.67 | 1.06 | 18.11 |
| DeepSeek-v3-0324 | 0.14 | 0.57 | 20.72 |
| GLM-4.5 | 0.14 | 0.99 | 13.35 |
| GLM-4.5-air | 0.14 | 1.31 | 15.18 |
| Grok-4 | 0.18 | 1.21 | 20.75 |
| Doubao-1.6 | 0.99 | 1.74 | 14.09 |

## F  COMPATIBILITY WITH SPECULATIVE DECODING

The Language Confusion Gate (LCG) is designed to be a lightweight module that integrates seamlessly with modern inference optimizations, including speculative decoding. This appendix details why LCG is compatible with this technique and does not compromise its efficiency gains.

**Speculative Decoding Overview.** Speculative decoding accelerates inference by using a smaller, faster "draft" model to generate a candidate sequence of several tokens. The larger, more powerful "target" model then processes this entire sequence in a single, parallel forward pass to verify the draft tokens. This allows multiple tokens to be accepted for the cost of a single forward pass of the large model, significantly improving throughput.

**Integration of LCG.** LCG's compatibility stems from its ability to be integrated directly into the target model's verification step with minimal overhead. The process works as follows:

1. **Draft Generation**: The small draft model generates a candidate sequence of tokens as it normally would.
2. **Target Verification**: The large target model performs its single forward pass on the input combined with the draft sequence, producing hidden states and logits for each token position in parallel.
3. **LCG Masking**: At each position, the LCG takes the corresponding hidden state generated by the target model and computes a language mask. Because the LCG is a small MLP, this step is extremely fast and can be performed in parallel for all draft tokens.
4. **Logit Modification**: The LCG's mask is applied to the target model's logits for each token. Since intervention is rare, the logits are often left unchanged, ensuring minimal impact on the model's standard behavior.
5. **Final Validation**: The standard speculative decoding algorithm proceeds with its validation check, but it uses the LCG-modified logits to accept or reject the draft tokens.

Because the LCG is a fast, parallelizable operation that adds negligible computational cost to the target model's pass, it does not create a bottleneck or negate the speed improvements offered by speculative decoding. This ensures that language confusion can be mitigated without sacrificing inference performance.

## G INVESTIGATION OF HOW TRAINING DATA AFFECTS OUTPUT TOKEN NORM

As discussed in Section 3.2, the imbalance of output token embedding norms contributes to language confusion. We studied how training data affects token norm in Olmo 3 (Team OLMo et al., 2024). Olmo 3 released their training data and checkpoint in different stages for research purpose. Here we compared the output token norm of Olmo-3-1025-7B (Pretrain), which is the pretraining checkpoint, and Olmo-3-7B-Instruct-SFT (SFT), which is the SFT checkpoint. According to Olmo 3's tech report, the training data is filtered in the pretraining stage to retrain contain English texts. While in the SFT stage, they incorporate the aya dataset (Singh et al., 2024) for multilingual capabilities. Since the percentage of low resource tokens in SFT stage is much higher than that of Pretrain stage (which is close to 0), we expect the low res token norm to increase.

In Table 8, we compare the percentage of tokens whose embedding norms lie in the top 5% of all token norms in the model's vocabulary of Olmo 3 Pretrain and Olmo 3 SFT, similar to Table 1. We can see that after the SFT stage, the percentage of Low-Res tokens with top 5% norm increase from 3.79% to 7.89%. Notice that since the pretraining stage of Olmo 3 only contains English, Chinese is also relatively "low resource" in this case. As a result, we observe that the percentage of CJ tokens with top 5% norm increases from 0.87% to 2.14% after the SFT stage.

Table 8: Percentage of tokens in each language family with embedding norms among top 5% of all token embedding norms in Olmo 3 7B pretrain and SFT stage.

| Model | CJ% | Latin% | Low-Res% |
|---|---|---|---|
| Olmo 3 7B Pretrain | 0.87 | 5.25 | 3.79 |
| Olmo 3 7B SFT | 2.14 | 5.22 | 7.89 |

Notice that the percentage of low-res tokens in in top 5% of all tokens norm in Olmo is higher than the 5 models we measured in Table 1. This is because Olmo only tokenize very high frequency

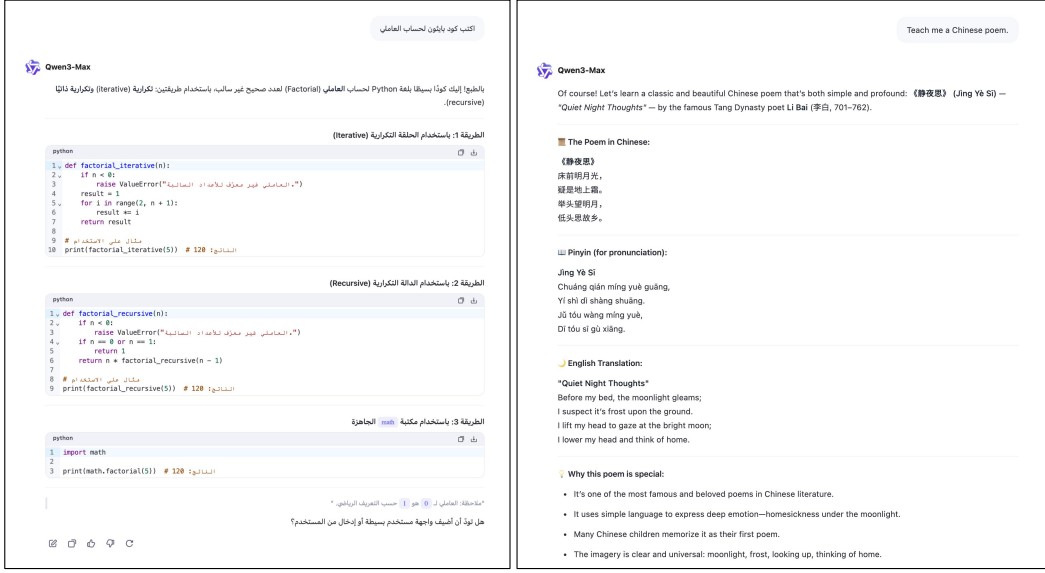

Figure 5: Left: User asking in Arabic to write a python program that computes factorial. Right: User asking in English to let LLM introduce a Chinese poem.

grams in low resource languages, which are mostly single characters. As a comparison, there are 19,257 Low-Res tokens in Qwen's tokenizer among of vocab size of 131,936, while only 1268 Low-Res tokens in Olmo 3's tokenizer among a vocab size of 100,278. Since Olmo 3 only tokenize high frequency Low-Res tokens, the average frequency of all Low-Res tokens are higher, which explains the relatively higher norm.

## H    EXAMPLES OF NATURAL CODE-SWITCH

We show examples of user queries that require LLM to naturally blend two languages in Figure 5.

## I    EXAMPLES OF INTERVENTION V.S. NO INTERVENTION

We show examples of Qwen3-8B responses to a query with and without LCG intervention in Figure 6. The example on the left shows an example that incorrect language mixing is avoided, and the example on the right shows an example that correct code-switch is maintained.

From Figure 7 to Figure 10. We show qualitative examples on Qwen3-30B-A3B-Instruct-2507 showing that models with LCG maintains natural code-switch capabilities under context that requries multilingual communication. For each of the Figures, on the left we show the response of Qwen3-30B-A3B-Instruct-2507 without intervention (NoGate), on the right we show the response of the same model with Intervention (Gate). We can see that model with LCG preserves capability to switch between languages.

## J    FALIURE CASES OF LCG

It's possible for LCG to fail in preventing language confusion. This happens when the LCG thinks token in a different language is acceptable here, while it turns out to be incorrect. We show two examples of this case in Figure 11.

However, it's impossible for LCG to increase rate of language confusion, because of the intervention rule of **"persistence of the previous token's language"** in Section 4.3. Language confusion can

**QUERY:**
Translate the following English text into Modern Standard Arabic. Only provide the translation. English: Layton had asked for changes to the conservatives' environmental bill during the meeting with the PM, asking for a "thorough and complete rewriting" of the Conservative party's environmental bill. Modern Standard
Arabic:

**Qwen3-8B No LCG:**
طلب ليتون إجراء تعديلات على قانون البيئة لحزب المحافظين خلال الاجتماع مع رئيس الوزراء، مطالباً بـ" إعادة كتابة شاملة وplete "لقانون البيئة لحزب المحافظين.

**Qwen3-8B LCG-adjusted:**
طلب ليتون إجراء تعديلات على قانون البيئة للحزب المحافظ خلال اللقاء مع رئيس الوزراء، مطالباً بـ"إعادة كتابة شاملة وشاملة" لقانون البيئة للحزب المحافظ.

**QUERY:**
Translate the following English text into Korean. Only provide the translation. English: Also, make sure to pronounce r and rr differently: caro means dear, whereas carro means chariot.
Korean:

**Qwen3-8B No LCG:**
또한, r와 rr를 다르게 발음하는 것을 확인하세요: caro는 사랑하는 분을, carro는 전차를 의미합니다.

**Qwen3-8B LCG-adjusted:**
또한, r와 rr를 다르게 발음해야 합니다: caro는 사랑하는 사람을 뜻하지만, carro는 전차를 뜻합니다.

Figure 6: Comparison between language confusion and natural code-switch.

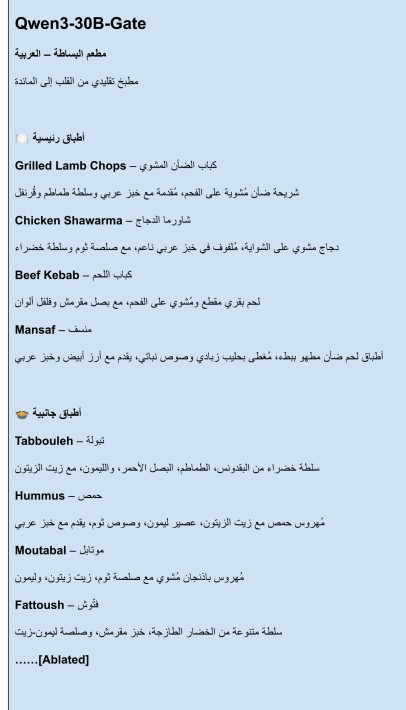

Figure 7: Qwen3-30B without and with LCG under multilingual context of Arabic and English.

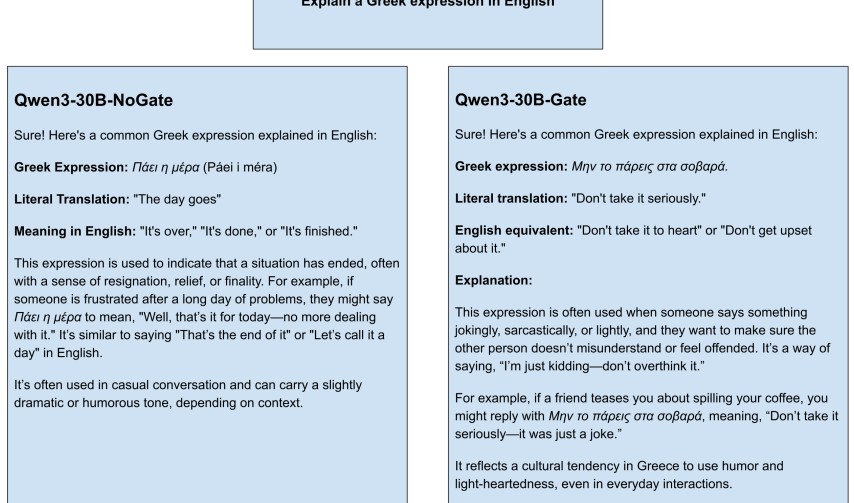

Figure 8: Qwen3-30B without and with LCG under multilingual context of Greek and English

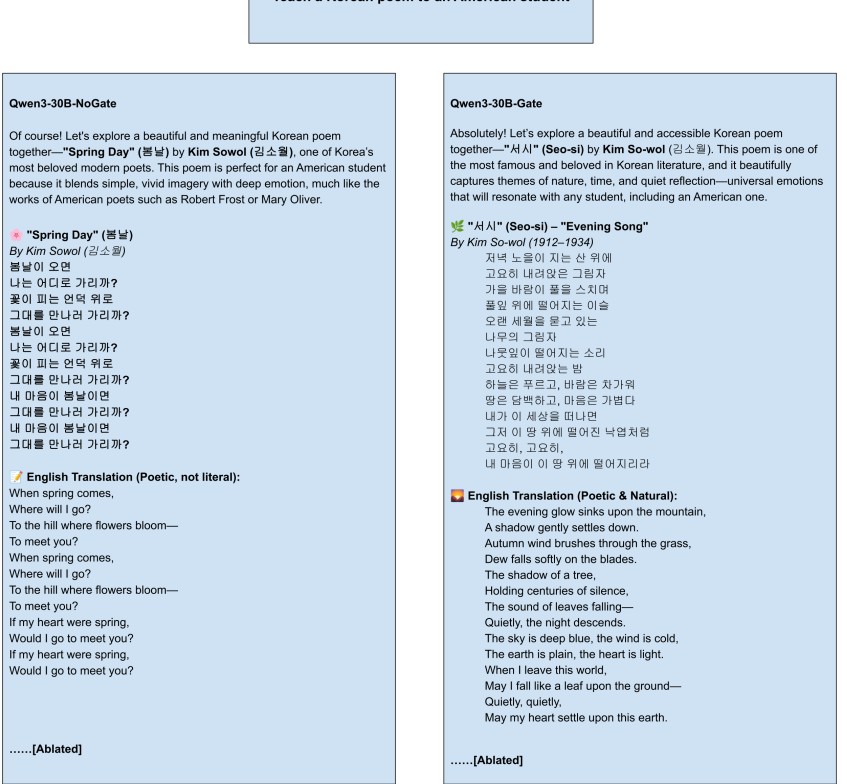

Figure 9: Qwen3-30B without and with LCG under multilingual context of Korean and English

**Write a dialogue between a Russian teacher and an American student**

**Qwen3-30B-NoGate**

**Title: A Conversation in the Classroom – Russian and English**

**Characters:**

- **Anna Petrova** – A Russian language teacher, mid-40s, warm but precise.
- **James Carter** – An American college student, early 20s, enthusiastic but struggling with pronunciation.

**[Setting: A small university classroom. A whiteboard has "Добро пожаловать!" (Welcome!) written in Cyrillic. James sits at a desk, nervously flipping through his notebook.]**

**Anna:** (smiling) Good morning, James! How are you today?

**James:** (hesitant) Uh… good morning, Ms. Petrova! I'm… um… okay. A little nervous.

**Anna:** (chuckles) Don't be nervous! We're here to learn, not to be perfect. Remember, even Russians make mistakes when speaking English.

**James:** (laughs) Yeah, I've seen that. My roommate says "I am very good at to speak English" all the time.

**Anna:** (grinning) Ah, yes! That's a classic! We say "I am good at speaking English." But don't worry—your Russian will get better too. Let's start with something simple. Today, we're learning greetings.

**James:** Great! I know "привет" and "здравствуйте." I've been practicing.

**Anna:** Excellent! "Привет" is informal—like "hi" to a friend. "Здравствуйте" is formal—like "hello" to a teacher or stranger. Now, how do you say "How are you?" in Russian?

**James:** (thinking) Uh… "Как дела?"

**……[Ablated]**

**Qwen3-30B-Gate**

**Title: "A Conversation Across Cultures"**

**Setting:** A cozy classroom at a university in Washington, D.C. A Russian teacher, **Irina Petrova**, is reviewing a lesson with her American student, **Jake Thompson**, after class. The room is filled with books, maps, and a small Russian nesting doll on the desk.

**Irina:** (smiling warmly) Jake, you've been studying Russian for nearly a year now. I'm proud of your progress. But I noticed something in your last assignment—your pronunciation of "привет" was a little off.

**Jake:** (laughing) Oh, I'm sorry, Professor Petrova. I've been practicing, but I swear I've been saying "privet" like you do. Maybe I just need more coffee?

**Irina:** (chuckling) Perhaps! But no, the issue is not your effort. It's the stress. In Russian, "привет" is pronounced with a soft, almost whispery "v" sound—like "pree-vyet," not "privet." Try it with me: *Pree-vyet.*

**Jake:** (trying) Pree-vyet…? Like a whisper?

**Irina:** Exactly! Like a secret. (She smiles.) You're American—so you're used to strong, clear sounds. But Russian is more… intimate. It's like speaking to a close friend in a quiet room.

**Jake:** That's actually kind of beautiful. I never thought of it that way. In English, we say "hello" loud and fast. But "привет" feels… personal.

**Irina:** (nodding) Yes! And that's the magic of language. It's not just words—it's feeling. When we say "спасибо" (spasibo), it's not just "thank you." It's gratitude with soul.

**Jake:** I like that. I've been reading *War and Peace*—well, the parts I can understand—and I keep thinking about how Tolstoy describes emotions so deeply. It's like reading poetry.

**……[Ablated]**

Figure 10: Qwen3-30B without and with LCG under multilingual context of Russian and English.

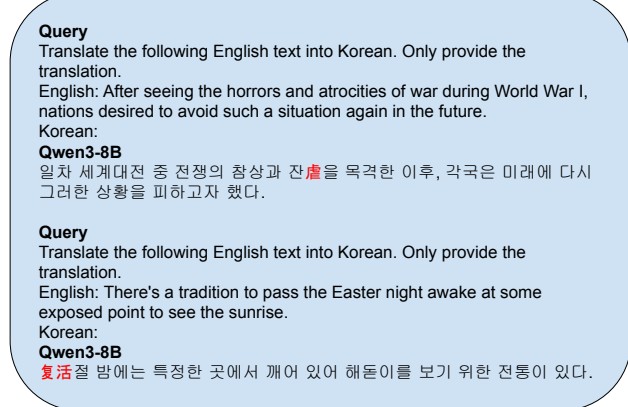

Figure 11: Examples of LCG fail to prevent language confusion on Qwen3-8B.

only happens when the language of a token is different from the last token, while the rule ensures that the probability of tokens with same language as last token would never decrease.

