# OpenReview forum: "Language Confusion Gate: Language-Aware Decoding Through Model Self-Distillation"
_ICLR.cc/2026/Conference — ICLR 2026 Poster_

### Official Review · Reviewer_EcN8 · 2025-10-31

**Soundness:** 4
**Presentation:** 3
**Contribution:** 3
**Rating:** 8
**Confidence:** 4

**Summary:**

This paper introduces the Language Confusion Gate, which is a mechanism to prevent language confusion. Language confusion is the scenario in which a model mistakenly uses tokens from an inappropriate language - such as including Latin text in a language that does not typically have code switching (e.g. Chinese or Japanese). This paper distinguishes between "harmful confusion and acceptable code-switching." The authors carefully construct test scenarios by analyzing token distributions, properties of unicode, and task relevant exceptions such as english technical terms in otherwise non-english text. The authors construct a classifier trained on final hidden states, accounting for interesting observations about HRL token norms, and use this classifier to gate allowed tokens at model decoding. This method is effective at reducing harmful confusion, permitting intentional language mixing, and maintains task performance. I found the analysis of why language confusion happens (top logit distributions, insufficiency of greedy decoding, high norm bias) particularly interesting and insightful.

**Strengths:**

This paper identifies a rare, but important, failure mode for LLMs. They provide interesting analysis to motivate studying and solving this problem, then introduce several specific technical innovations to address it.

S1. This paper is extremely well motivated. I was initial skeptical of the prevalence of the problem, but the paper expands on prior work by Marchisio et al. showing that the problem setting is realistic. Sections 2 and 3 are particularly effective in exploring why harmful language confusion happens (e.g. showing that the language confusion token is the top-1 choice 56% time).

S2. Paper introduces high resource language (HRL) norm bias. The authors show that output tokens with very high norm are naturally higher in the distribution (230-241), and that HRL tokens are over-represented in the top 5% of token output norms. In 264-265 the authors are very precise about the claims, stating they do not claim that this phenomena accounts for all confusion and thus cannot be used as a direct signal. Rather they use this to normalize measurements when training the classifiers.

S3. The classifier and gating mechanism are very carefully designed - section 4.3 lists deterministic rules of when the classifier can override the expected token prediction. These ensure that the masking does not prevent fluent outputs.

S4. The evaluations are broad and relevant, including both translation focused tasks derived from FLORES and tasks suitable for reasoning models. The evaluations also carefully distinguish between harmful and intentional code mixing (e.g. WITH-LATIN scenario).

S5. Appendix A details the analysis of partial unicode characters present in model vocabulary - these are details often missed in other papers that study tokenization. The examples in the appendix and discussion of connections between token gating and speculative decoding are also interesting.

Overall this is a very strong and well-executed paper.

**Weaknesses:**

None of these weaknesses should be taken as a reason to avoid publication.

- It would be interesting to see how experimental results change if the additional deterministic intervention rules were not used to further control the gating mechanism

- If there exist a lightweight way to do experiments going beyond the CJ/Latin split, it would be quite interesting to see those results. Perhaps leveraging further unicode metadata? However this is not necessary, it would simply further strengthen the paper.

- Further investigation of the high-norm tokens would also be interesting. Could these be tied to token frequencies in pretraining data, for models with known data (e.g. olmo models?). Also, the causality direction between high norms <> HRL tokens could be further investigated.

**Questions:**

- Could you provide statistics indicating how many tokens are disallowed at certain timestamps?

Line 206 contains a typo

---

> ### Author Response · Authors · 2025-11-24
>
> > **Ablation of intervention rules**
>
> Thank you for the suggestion. We agree that an ablation study is important. We conducted experiments on Qwen3-8B and Llama3.1-8B:
>
> | Metric | Qwen3-8B | Llama3.1-8B |
> | :--- | :--- | :--- |
> | **CJ Confusion (No LCG)** | 4.47 | 3.01 |
> | **CJ Confusion (No Rules)** | 1.42 | 0.46 |
> | **CJ Confusion (LCG-adjusted)** | 0.11 | 0.28 |
> | **Latin Confusion (No LCG)** | 12.09 | 8.40 |
> | **Latin Confusion (No Rules)** | 5.07 | 3.90 |
> | **Latin Confusion (LCG-adjusted)** | 2.41 | 2.77 |
>
> Removing the rules decreases performance compared to the full LCG method, though it remains better than the baseline. This demonstrates the necessity of both the LCG mechanism and the intervention rules. We added these numbers to Figure 3 and discussed them in Section 5.3.
>
> > **Go beyond CJ/Latin split**
>
> We acknowledge that our mechanism uses broad families. However, our observations indicate that confusion generally happens from low-resource languages to high-resource ones (English/Chinese). Consequently, LCG eliminates the vast majority of confusion cases. We discuss this in the Conclusion. In future work, we aim to add granularity to the Low-Res category. However, distinguishing between languages that share the same script at the token level is impossible because of the overlap of tokens in different languages.

---

> ### Author Response · Authors · 2025-11-24
>
> > **Further investigation of the high-norm tokens**
>
> Studying how training data affects token output norm embeddings is an interesting direction. While a comprehensive causal analysis is difficult since other factors like regularization and optimizers can also affect output token norm, we performed a preliminary analysis on Olmo 3 under a controlled setup.
>
> Olmo-3 filters data to be English-only during pre-training, while the post-training (SFT) stage integrates the multilingual Aya dataset. We compared the **Pretrain checkpoint** (allenai/Olmo-3-1025-7B) with the **SFT checkpoint** (allenai/Olmo-3-7B-Instruct-SFT). We examined the percentage of tokens within the top 5% of norms for each category:
>
> * **Pretrain:** CJ (0.87%), Latin (5.25%), Special (1.48%), Low-Res (3.79%)
> * **SFT:** CJ (2.14%), Latin (5.22%), Special (2.26%), **Low-Res (7.89%)**
>
> We can see that after the SFT stage, the percentage of Low-Res tokens in the top 5% of norms increased significantly. This is because the percentage of low resource tokens in pretraining data is low, while higher in SFT stage because of the integration of the multilingual Aya dataset. This shows that training with more Low-Res tokens increase Low-Res token norms. **We have added this discussion to Appendix G.**
>
> It is worth noting that Olmo 3 has a higher percentage of high-norm Low-Res tokens compared to other models like Qwen 3 as shown in Table 1 of our revision pdf. This is due to the different tokenization/vocabulary of Olmo 3:
> * Qwen: 19,257 Low-Res tokens among vocab size of 131,936 (often multi-character).
> * Olmo-3: 1,268 Low-Res tokens among vocab size of 100,278 (mostly single-character).
>
> Because Olmo-3 tokenizes low-resource languages primarily as single characters, these tokens have higher relative frequencies, contributing to higher norms. Since Olmo 3 only tokenize high frequency Low-Res tokens, the average frequency of all Low-Res tokens are higher, this explains the relatively higher percentage of high-norm Low-Res tokens.
>
> Here’s an example of qwen’s tokenization of an Arabic sentence vs Olmo3, we tokenize the sentence with the two tokenizers, and convert tokens back to text to inspect the length of each token.
>
> ```
> text = 'اللغة العربية تُعد من أقدم اللغات الحية في العالم، وتتميز بثراء مفرداتها وجمال أسلوبها، مما يجعلها وسيلة تعبير قوية في الأدب والشعر والخطاب الديني.'
> # Translation: Arabic is one of the oldest living languages in the world, distinguished by the richness of its vocabulary and the beauty of its style, making it a powerful medium of expression in literature, poetry, and religious discourse.
> Length of Qwen tokens: 55
> Qwen tokens:
> ['ال', 'لغ', 'ة', ' العربية', ' ت', 'ُ', 'عد', ' من', ' أ', 'قدم', ' اللغ', 'ات', ' الح', 'ية', ' في', ' العالم', '،', ' وت', 'تميز', ' ب', 'ثر', 'اء', ' م', 'فرد', 'اتها', ' و', 'جمال', ' أ', 'سل', 'وب', 'ها', '،', ' مما', ' يجعل', 'ها', ' و', 'س', 'يلة', ' ت', 'عب', 'ير', ' ق', 'وية', ' في', ' الأ', 'دب', ' وال', 'شعر', ' وال', 'خط', 'اب', ' ال', 'دي', 'ني', '.']
> Length of Olmo tokens: 100
> Olmo tokens:
> ['ال', 'ل', 'غ', 'ة', ' ال', 'ع', 'ر', 'ب', 'ية', ' ت', 'ُ', 'ع', 'د', ' من', ' أ', 'ق', 'د', 'م', ' ال', 'ل', 'غ', 'ات', ' ال', 'ح', 'ية', ' في', ' ال', 'ع', 'ال', 'م', '،', ' و', 'ت', 'ت', 'م', 'ي', 'ز', ' ب', 'ث', 'ر', 'اء', ' م', 'ف', 'ر', 'د', 'ات', 'ه', 'ا', ' و', 'ج', 'م', 'ال', ' أ', 'س', 'ل', 'و', 'ب', 'ه', 'ا', '،', ' م', 'م', 'ا', ' ي', 'ج', 'ع', 'ل', 'ه', 'ا', ' و', 'س', 'يل', 'ة', ' ت', 'ع', 'ب', 'ير', ' ق', 'و', 'ية', ' في', ' ال', 'أ', 'د', 'ب', ' و', 'ال', 'ش', 'ع', 'ر', ' و', 'ال', 'خ', 'ط', 'اب', ' ال', 'دي', 'ن', 'ي', '.']
> ```
>
> We can see that Olmo3 tokens are mostly single character, while qwen tokens are multi character, thus qwen only use 55 tokens, while olmo3 use 100 tokens. This shows that Olmo3 only tokenize low res language tokens with highest frequencies (which are mostly single characters). As a result, the frequency would be higher than multi-characters.
>
> > **Percentage of tokens disallowed by intervention**
>
> We agree that intervention statistics help contextualize the effect of LCG. We measured intervention rate at token level on the FLORES-NO-LATIN dataset:
> * **Qwen3-8B:** 0.38% (523 of 139,354 tokens).
> * **Llama3.1-8B:** 0.33% (545 of 162,846 tokens).
>
> This demonstrates that LCG intervention is sparse and precise, having minimal influence on normal token generation. We added these statistics to Section 5.3 in the Frequency of Intervention paragraph.

---

### Official Review · Reviewer_53Zx · 2025-11-01

**Soundness:** 3
**Presentation:** 2
**Contribution:** 2
**Rating:** 6
**Confidence:** 3

**Summary:**

The paper proposes the Language Confusion Gate (LCG), injecting a lightweight MLP layer into LLM to reduce unintended mixing of scripts/languages during LLM decoding. This is based on the observations (i) language confusion is rare and the correct-script token is usually ranked in the top-k, (ii) embedding-norm imbalance favors high-resource scripts. The MLP layer is trained to predict which language families should be permitted at each step and mask others in the logits LCG reduces both Chinese/Japanese and Latin confusion by up to an order of magnitude on translation and QA settings (FLORES+, INCLUDE) and on “thinking” models for code generation (HumanEval-XL), with negligible or no degradation in BLEU/accuracy/Pass@k. This method works for both thinking and non-thinking models.

**Strengths:**

1. The paper introduces a norm-adjusted self-distillation training signal for a language-family gate. I consider it as a very simple idea distinct from weight edits or reward finetuning.

2. The paper demonstrates empirical gains of the proposed method across open and closed models.

**Weaknesses:**

1. The paper calculates “code-switch rate” on FLORES-WITH-LATIN but does not include human judgments or llm-as-a-judge on the responses. It could offer more evaluation on whether the improved responses are also preferred by humans.

2. Are there cases that the proposed method do not help to correct the language confusion and could even increase the problem? It would be helpful to showcase some negative cases as well.

**Questions:**

1. How do you perform the threshold calibration for the sigmoid function in the MLP? What is the criteria to determine that?
2. Do you try any multilingual tasks, where the model need to respond in at least two different languages? Does the proposed method help in that scenarios?

---

> ### Author Response · Authors · 2025-11-24
>
> > **LLM-as-a-judge eval for FLORES-WITH-LATIN**
>
> Thank you for the suggestion. We added an evaluation using GPT-4 as a judge for our FLORES-WITH-LATIN experiments to measure code-switching rates. The results below show the win rate (how often the LCG version is preferred over the baseline):
>
> * **Qwen3-8B:** Win: 48.01%, Tie: 6.84%, Loss: 45.11%
> * **Llama3-8B:** Win: 48.86%, Tie: 2.85%, Loss: 48.29%
> * **Gemma-12B:** Win: 34.53%, Tie: 32.90%, Loss: 32.57%
>
> The gate slightly improves the results. This is expected, as language confusion (and thus intervention) occurs sparsely.
>
> > **Negative cases and possibility of increasing confusion**
>
> It is valuable to analyze cases where LCG does not prevent confusion. We have added the following examples to the revision:
>
> ```
> **Query**
> Translate the following English text into Korean. Only provide the
> translation.
> English: After seeing the horrors and atrocities of war during World War I,
> nations desired to avoid such a situation again in the future.
> Korean:
> **Qwen3-8B LCG**
> 일차 세계대전 중 전쟁의 참상과 잔虐을 목격한 이후, 각국은 미래에 다시
> 그러한 상황을 피하고자 했다.
> (虐 is language confusion)
>
> **Query**
> Translate the following English text into Korean. Only provide the
> translation.
> English: There's a tradition to pass the Easter night awake at some
> exposed point to see the sunrise.
> Korean:
> **Qwen3-8B LCG**
> 复活절 밤에는 특정한 곳에서 깨어 있어 해돋이를 보기 위한 전통이 있다.
> (复活 is language confusion)
> ```
>
> Intervention fails when: (1) the gate incorrectly identifies the token as a natural code-switch, or (2) the model is so confident in the incorrect language token that no valid language-consistent token exists within the top-p/top-k sampling pool.
>
> However, we argue that LCG cannot worsen the problem. The third rule in Section 4.3 ("Persistence of the previous token’s language") ensures that the probability of a token matching the previous language does not decrease. Thus, LCG cannot strictly increase confusion. We have added this discussion to Appendix I.
>
> > **MLP sigmoid threshold**
>
> Thank you for noting this. We did not specifically calibrate the threshold; we set it to the default of 0.5.
>
> > **Multilingual tasks**
>
> One typical multilingual task with a reliable evaluation metric involves coding: the user asks in a low-resource language, and the LLM explains in that language while writing code in Latin script. This is evaluated using the HumanEval-XL benchmark (Table 6), where the model must switch languages to perform well.
>
> For broader multilingual contexts, evaluation is more challenging. We provided four qualitative examples of queries requiring multi-language responses on Qwen3-30B-A3B. These show that LCG preserves the model’s code-switching ability. We have added these examples and discussion to Appendix J.
>
> We list the four prompts we tested below, please refer the revision pdf for the comparison of model responses:
> - Teach a Korean poem to an American student
> - Create an Arabic restaurant menu with Arabic text and English dish names
> - Explain a Greek expression in English
> - Write a dialogue between a Russian teacher and an American student

---

### Official Review · Reviewer_KkFU · 2025-11-01

**Soundness:** 3
**Presentation:** 3
**Contribution:** 3
**Rating:** 6
**Confidence:** 3

**Summary:**

This paper addresses the language confusion issue in LLMs, where models inappropriately mix languages during generation. It proposes the LCG (Language Confusion Gate), a lightweight plug-in module that filters tokens during decoding without modifying the base LLM. Trained via norm-adjusted self-distillation, LCG predicts possible language families and applies masking when necessary. Experiments across various models show that LCG reduces language confusion while maintaining task performance.

**Strengths:**

1. The proposed LCG is a practical and lightweight solution, avoiding the need for model retraining and adding minimal computational overhead.

2. The norm-adjusted self-distillation method effectively mitigates the bias towards high-resource languages caused by token embedding norm imbalance.

3. The work distinguishes between harmful language confusion and legitimate code-switching, ensuring the model retains necessary multilingual expression capabilities.

**Weaknesses:**

1. LCG only classifies tokens into broad language families (CJ, Latin, Symbols, Low-Res), failing to resolve confusion between languages sharing the same script (e.g., Chinese vs. Japanese).

2. The evaluation’s reliance on rule-based detectors for language confusion may have limitations, especially in complex multilingual contexts.

3. The paper does not discuss the generalization of LCG to more low-resource languages beyond the tested ones.

**Questions:**

1. Could the token classification method be optimized to handle more fine-grained language distinctions rather than just broad families?

2. What is the specific computational overhead of LCG in large-scale inference?

---

> ### Author Response · Authors · 2025-11-24
>
> > **Concern on broad families**
>
> We acknowledge that our intervention mechanism operates on broad language families. However, we observed that language confusion mostly happens when the LLM mixes high-resource languages (English, Chinese) to low-resource languages (e.g., Arabic, Korean). Therefore, LCG is able to eliminate the majority of confusion cases, decreasing the language confusion rate by an order of magnitude as shown in our experiments. We have discussed this limitation in the Discussion and Conclusion sections.
>
> > **Reliance on rule-based detectors**
>
> We agree that rule-based detectors have limitations; however, we maintain that:
> 1.  **Context-Specific Validity:** Our evaluation combines carefully curated datasets with heuristic detectors. We do not claim these detectors work universally, but they are robust within our specific experimental setup. For example, in the INCLUDE and HumanEval datasets, we only evaluate CJ (Chinese/Japanese) confusion, since the prompts do not require CJ tokens, a rule-based detector is reliable. For the FLORES dataset, we designed an automated method to select samples that should not contain Latin characters by inspecting reference answers, as detailed in Section 5.2.
> 2.  **Best Available Automated Proxy:** We believe this approach ("curated dataset + rule-based detector") is the best available method for evaluating language confusion without manual human labeling. An LLM-based detector is unsuitable because, as our experiments show, LLMs themselves suffer from language confusion.
>
> > **Generalization of LCG to more low-resource languages**
>
> We argue that our current experiments already demonstrate strong generalization:
> * We cover a wide range of languages of Arabic, Hebrew, Korean, Thai, Greek, Russian, and Vietnamese; for some of the benchmarks, we are limited in testing more because standard benchmarks cover a finite set.
> * The evaluation datasets differ from the training datasets, ensuring we have not "optimized" for the evaluation set.
> * Our training dataset covers 200+ languages, suggesting that the current performance should extrapolate well to other languages.
>
> > **Computational overhead of LCG**
>
> We agree that quantifying inference speed is important. We implemented efficient kernels in vLLM to support LCG. In our experiments on Qwen-30B-A3B-Instruct-2507 (8 concurrency, 2k input tokens, 100 token output), the time per token is 15.94ms without the gate and 15.99ms with the gate. This is an increase of 60us (0.4%), which is negligible. We have added this to the Discussion section.

---

### Official Review · Reviewer_7Ns2 · 2025-11-03

**Soundness:** 3
**Presentation:** 4
**Contribution:** 3
**Rating:** 8
**Confidence:** 3

**Summary:**

Authors propose language confusion gate (LCG), which uses a 2-layer MLP to predict the language class of next generated tokens and mask the unwanted language tokens. Results show that language confusion behaviors are reduced by 10x by applying the proposed method.

**Strengths:**

1. Authors propose an effective fix on the problem of language confusion based on reliable observations.
2. Method is simple and intuitive.
3. That the method does not require modifying the base model alleviates the concern of the potential forgetting happening.

**Weaknesses:**

1. In section 5.4, there is slight performance degradation on Qwen series after adjusting for LCG (no degradation on GPT-OSS though). Further error analysis on this matter is needed. For example, some qualitative analysis on some reasoning traces: does LCG suppress the diversity/exploration in language models?
2. In light of the above, more experiments could be done evaluating general capabilities after applying LCG.

**Questions:**

1. Could authors measure if there is any effect on inference speed?
2. Is there potentially a cleaner way to obtain the expected class of the next token? For example rule-based/heuristics-based on logits/last hidden states. Would like to see such a simple baseline to be experimented with to justify a two-layer MLP is needed.
3. Does the incorporation of LCG affect the length of reasoning traces?

---

> ### Author Response · Authors · 2025-11-24
>
> > **Slight performance degradation on Qwen in Section 5.4**
>
> Thank you for pointing this out. We examined this issue carefully and found that the performance drop was due to our post-processing of the LLM response being too strict before feeding it into the HumanEval grader. After correcting the post-processing logic, the new results show that LCG has a negligible influence on performance. We have updated Table 4 in the revision to reflect these corrected results.
>
> > **More experiments on evaluating general capabilities**
>
> We agree that measuring the effects of LCG on general capabilities is crucial. We measured the impact of LCG on **Qwen-30B-A3B-Instruct-2507**, the most performant model in our experiments. We compared performance with and without LCG across a broad range of benchmarks: **MMLU-ProX-13, MMLU, MMLU-Pro, GPQA, SuperGPQA, AIME24, and AIME25**. The results indicate no statistically significant impact on general capabilities. We have added this table to Appendix B of the revised paper.
>
> > **Effect on inference speed**
>
> We agree that measuring the effect on inference speed clarifies the practical impact of LCG. To support LCG in production systems, we implemented efficient kernels in vLLM. In our experiments on **Qwen-30B-A3B-Instruct-2507** (with 8 concurrency, 2k input tokens, and 100 token output), the time per token is **15.94ms without the gate** and **15.99ms with the gate**. This represents an increase of only 60us (0.4%), which is negligible. We have added these details to Section 6 (Discussion).
>
> > **Rule-based baseline**
>
> As discussed in Section 3.3, it is necessary to distinguish between *language confusion* and *natural code-switching*. Rule-based or heuristic detectors lack the linguistic understanding in comparison to model-based intervention mechanisms. In Appendix I, we added examples of contexts requiring the LLM to switch between languages. A simple rule-based intervention mechanism would restrict the model to a single language, effectively removing its multilingual capabilities.
>
> > **Effect on reasoning trace length**
>
> Thank you for the suggestion. We measured the effect of LCG on reasoning trace length using HumanEval for the experimented models.
> * **Qwen3-8B:** Increased from 3327 to 3361 tokens.
> * **GPT-OSS-20B:** Increased from 501 to 507 tokens.
> * **30B-Think:** Increased from 2528 to 2534 tokens.
>
> We observe that LCG increases the reasoning chain by only a few tokens, making the effect minimal. We have added these findings to Table 4.

---

> > ### Comment · Reviewer_7Ns2 · 2025-11-27
> >
> > Thanks authors for the additional experiments. I will maintain my original positive score.

---

### Author Response · Authors · 2025-11-24

Here we summarize the changes we made to the revision pdf.

We have incorporated the following changes based on the reviewers’ suggestions:

* **Data Corrections:** We updated Table 4 with fixed numbers and reasoning length data.
* **General Capabilities Benchmarking:** We benchmarked the effect of LCG on Qwen3-30B-A3B across a broader range of datasets to demonstrate that LCG does not negatively impact the model’s general performance.
* **Inference Overhead:** We added a discussion in Section 6 regarding the minimal overhead introduced by LCG during inference.
* **Failure Cases:** We discussed potential failure cases of LCG in Section J and provided two specific examples.
* **Qualitative Analysis:** We added more qualitative examples in Appendix I showing that code-switching and multilingual conversation capabilities are preserved.
* **Ablation Study:** We added an ablation study of the intervention rules in Figure 3.
* **Token Norm Analysis:** We performed a case study on Olmo 3 in Appendix G to analyze how the percentage of low-resource tokens affects the token norm.

In addition to these points, we have made the following improvements to the paper:

* **Typo Fixing** We fixed the typos found in the original text.
* **Structural Improvements:** We moved the discussion of token norm imbalance from Section 4 to Section 3 and improved the writing for better clarity. We also moved Figure 2 from the Appendix to Section 3.2, improving its design to make it easier to interpret.
* **Baseline Comparison:** We added a comparison with a training-based baseline, ORPO, in Figure 3. We find that LCG outperforms the training-based baseline on reducing language confusion without affecting the LLM’s general performance (evidenced by the degradation of the INCLUDE score in the ORPO baseline). This highlights the advantage of a mechanism that does not alter the model’s parameters.

---

### Meta-Review · Area_Chair_fCSi · 2026-01-09

**Summary:**

Authors propose an effective fix on the problem of language confusion based on reliable observations. That the method does not require modifying the base model alleviates the concern of the potential forgetting happening. The simple and effective idea is also practical. IT solves a real problem with lightweight solutions.

**Reviewer Concerns:**

The following concerns have been adequately addressed:
- reasoning length data is missing in experimental results.
- General Capabilities Benchmarking is missing for LCG.
- Inference Overhead: We added a discussion in Section 6 regarding the minimal overhead introduced by LCG during inference.
- Lack discussion on potential failure cases of LCG.
- Qualitative examples are missing for showing that code-switching and multilingual conversation capabilities are preserved.
- Ablation study is missing.
- how does the percentage of low-resource tokens affects the token norm?

**Reviewer Scores:**

The reviewers would have kept their high scores.

---

### Decision · Program_Chairs · 2026-01-26

Accept (Poster)